# Coagulation Protease-Driven Cancer Immune Evasion: Potential Targets for Cancer Immunotherapy

**DOI:** 10.3390/cancers16081568

**Published:** 2024-04-19

**Authors:** Subhojit Paul, Tanmoy Mukherjee, Kaushik Das

**Affiliations:** 1School of Biological Sciences, Indian Association for the Cultivation of Science, Jadavpur, Kolkata 700032, West Bengal, India; bcsp3@iacs.res.in; 2Department of Cellular and Molecular Biology, The University of Texas at Tyler Health Science Center, Tyler, TX 75708, USA; tanmoy.mukherjee@uttyler.edu; 3Biotechnology Research and Innovation Council-National Institute of Biomedical Genomics, Kalyani 741251, West Bengal, India

**Keywords:** biomarker, blood coagulation, cancer, clinical trial, coagulation protease, immune evasion, immunotherapy, protease-activated receptor

## Abstract

**Simple Summary:**

Blood coagulation and cancer are intrinsically related; hyper-thrombotic complications are often observed in certain types of tumors. On the other hand, coagulation proteases are shown to promote the progression of cancer in a variety of unique mechanisms. The present review highlights how coagulation protease-induced signaling contributes to the immune evasion of different cancers, leading to the development and progression of cancer. Understanding the mechanistic insights will aid developing novel therapeutics against coagulation protease-driven response, which could be used in combination with conventional immune checkpoint inhibitors to increase the efficacy of cancer treatment and patient survival.

**Abstract:**

Blood coagulation and cancer are intrinsically connected, hypercoagulation-associated thrombotic complications are commonly observed in certain types of cancer, often leading to decreased survival in cancer patients. Apart from the common role in coagulation, coagulation proteases often trigger intracellular signaling in various cancers via the activation of a G protein-coupled receptor superfamily protease: protease-activated receptors (PARs). Although the role of PARs is well-established in the development and progression of certain types of cancer, their impact on cancer immune response is only just emerging. The present review highlights how coagulation protease-driven PAR signaling plays a key role in modulating innate and adaptive immune responses. This is followed by a detailed discussion on the contribution of coagulation protease-induced signaling in cancer immune evasion, thereby supporting the growth and development of certain tumors. A special section of the review demonstrates the role of coagulation proteases, thrombin, factor VIIa, and factor Xa in cancer immune evasion. Targeting coagulation protease-induced signaling might be a potential therapeutic strategy to boost the immune surveillance mechanism of a host fighting against cancer, thereby augmenting the clinical consequences of targeted immunotherapeutic regimens.

## 1. Introduction

Blood coagulation is a tightly regulated biological process that prevents excessive bleeding upon injury to a blood vessel [1]. The process is facilitated by the conversion of the liquid form of blood into a semisolid mass, thereby forming a clot [2]. Vessel injury triggers the activation of the blood coagulation cascade, in which several clotting proteases function in orchestration to cease profuse bleeding [3]. Apart from their common role in coagulation, clotting proteases also induce cellular signaling via the activation of a G protein-coupled receptor (GPCR) family of proteases called protease-activated receptors (PARs) [4,5]. Blood coagulation and cancer are intrinsically related; enhanced blood coagulation is often observed in certain cancer patients, known as cancer-associated thrombosis (CAT), which is considered to be the second leading cause of mortality in cancer patients [6,7]. Moreover, the expression of PARs is shown to be significantly up-regulated in different types of solid tumors [8,9], and coagulation protease-driven PAR signaling often plays a crucial role in the progression of cancer [10,11]. PAR signaling not only supports tumor growth [12] but also promotes cancer metastasis [11]. Furthermore, the activation of PARs has also been shown to promote tumor angiogenesis [13]. Additional studies indicate that PAR activation induces epithelial-to-mesenchymal transition (EMT) in certain types of cancer, facilitating tumor metastasis [14]. EMT is characterized by an up-regulation of immune-suppressing mechanisms, leading to a futile anti-tumor immune response, ultimately contributing to the immune evasion of cancer [15]. Although the contribution of coagulation proteases and their receptors are well-established in tumor growth, metastasis, and angiogenesis, their role in cancer immune evasion occurs only at the initial stages. The present review thoroughly summarizes how the coagulation protease-mediated activation of PARs influences cancer immune evasion. The first part of the review highlights how different coagulation proteases activate different PAR isoforms. This is followed by a brief understanding of the role of PARs in tumor progression. We also briefly focus on the immune evasion mechanisms facilitating tumor survival and progression. The latter part of the review highlights how different coagulation proteases contribute to the immune evasion mechanisms of certain cancers. Finally, we discuss the recent immunotherapy-based clinical trials for different types of cancer.

## 2. PAR Activation by Different Coagulation Proteases

Mammalian cells express four PAR isoforms: PAR1, PAR2, PAR3, and PAR4. In 1991, PAR1, the prototypical receptor that is encoded by the *F2R* gene [16,17], was discovered on human platelets as the receptor of coagulation protease thrombin. Thereafter, other PAR family members, PAR2, PAR3, and PAR4, which are encoded by *F2RL1* [18], *F2RL2* [19], and *F2RL3* [20], respectively, were discovered through homology screening [20,21,22,23,24]. An irreversible proteolytic mechanism triggers the activation of PARs. Coagulation proteases bind to and cleave PARs at the N-terminal end, exposing a new N-terminal peptide that serves as a tether ligand and promotes PAR activation upon binding to a conserved region of extracellular loop 2 (ECL2) on the receptor (Figure 1) [16,25]. Therefore, synthetic peptides mimicking the first six amino acids of the newly generated N-terminus of PARs can activate the receptor in the absence of coagulation proteases [26,27,28]. However, an exception exists: PAR3 shows unresponsiveness towards synthetic agonist peptides [29]. The ligand activation of PARs induces a conformational change in the receptor, resulting in the affinity of the receptor to intracellular G proteins being altered, thereby triggering the signaling response [30]. Occasionally, PAR activation also triggers the phosphorylation of the receptor itself by G protein-coupled receptor kinase (GRK), aiding in the recruitment of β-arrestin, and GRK transduces signaling through it while preventing G protein-associated signaling [31,32]. However, the same PAR can be cleaved at different N-terminal sites by different coagulation proteases, leading to different conformational settings and associated signaling responses. This multiple, site-specific cleavage of a PAR by different coagulation proteases justifies how different signaling events are mediated by the activation of the same PAR [33,34,35]. The different coagulation protease-driven PAR signaling, and their specific cleavage sites are summarized in Table 1.

### 2.1. PAR1 Signaling via Coagulation Proteases

Coagulation protease thrombin binds to the N-terminal extracellular domain (exodomain) of PAR1 and cleaves the Arg41-Ser42 (R41-S42) peptide bond [36,37]. In addition to this, thrombin also binds to the acidic hirudin-like sequence of PAR1 at the C-terminus end [36,44]. The C-terminal end binding of PAR1 imparts the target specificity of thrombin. Thrombin-induced PAR1 cleavage promotes the interaction of the PAR1 C-terminal loop with G_q_ and G_12/13_ subfamilies of intracellular G-protein, leading to the signaling response [45,46,47]. Coagulation protease-activated protein C (aPC) activates PAR1 [48] via cleavage at the Arg46 (R46) residue of the exodomain [38,39]. Unlike thrombin, aPC does not interact with the hirudin-like domain of PAR1 [49], but it essentially requires binding with endothelial cell protein C receptor (EPCR) for PAR1 cleavage [50,51,52]. Instead of G proteins, aPC-driven PAR1 signaling involves intracellular β-arrestins, specifically β-arrestin-2 [53]. Clotting protease-activated factor VII (FVIIa) [54,55], the structural analog of aPC, also cleaves PAR1 depending on EPCR binding [56,57,58,59]. However, unlike aPC, FVIIa cleaves PAR1 at the R41 site and signals through β-arrestin-1 [35,60,61]. Like thrombin, activated factor X (FXa)-mediated PAR1 signaling requires cleavage at the canonical site, R41 [34,40], and is believed to be associated with intracellular G proteins [34,62]. However, the requirement of EPCR in FXa-mediated PAR1 signaling remains controversial. Some studies suggest the involvement of EPCR [63], whereas others indicate the EPCR-independent activation of PAR1 by FXa [64]. In contrast, other studies indicate that the FXa-driven activation of PAR1 is dependent on binding with a unique protein: annexin 2 [65]. Another coagulation-associated protease, plasmin, is shown to cleave PAR1 at multiple sites, K32-A33, R41-S42, R70-L71, K76-S77, and K82-Q83 [41]; however, the mechanism of plasmin action through PAR1 is yet to be established. Kallikrein-14, another coagulation-related serine protease, is also reported to activate PAR1 [66], probably via cleavage at the R46-N47 site [5].

### 2.2. Coagulation Protease-Driven PAR2 Signaling

Coagulation protease FVIIa is well-known for the cleavage and subsequent activation of PAR2. FVIIa binds to its primary receptor, tissue factor (TF), and transduces signaling via PAR2 [67,68], thereby exerting various cellular responses [69]. Similar to FVIIa, FXa also triggers PAR2 signaling [67,68]. Thrombin, on the other hand, at a very high concentration (100–500 nM), is shown to cleave PAR2, thereby activating the receptor [70]. Kallikrein-14 induces Ca^2+^ signaling in human embryonic kidney cells via the activation of PAR2 [66]. All these proteases preferentially cleave the peptide bond of PAR2 at R36-S37 [5,42,43].

### 2.3. PAR3 Signaling by Coagulation Proteases

A low thrombin concentration is shown to activate PAR3 [29]; however, thrombin-mediated PAR3 cleavage essentially requires the presence of PAR4 [29]. aPC-driven signaling in various cell types is also reported to be mediated by PAR3 [71,72,73]. Like aPC, FXa also triggers PAR3 activation and promotes signal transduction [74]. aPC and FXa preferentially cleave PAR3 at the R41-G42 site [5,39], whereas thrombin-induced PAR3 activation occurs via cleavage at K38-T39 [22].

### 2.4. PAR4 Activation via Coagulation Proteases

At a relatively high concentration, thrombin cleaves PAR4 at R47-G48 [30], thereby activating the receptor [75]. In some instances, Kallikrein-14 is also shown to induce signaling via the proteolytic cleavage of PAR4. However, the cleavage site for Kallikrein-14 remains unknown.

## 3. Coagulation Protease-Driven PAR Signaling in Cancer

Coagulation protease-mediated PAR signaling often plays a key role in the growth, development, and metastasis of tumors. The present section briefly highlights the contribution of different PARs to cancer progression. Figure 2 and Table 2 briefly summarize the role of coagulation protease-driven PAR signaling in cancer pathogenesis.

### 3.1. Coagulation Proteases and PAR1 Signaling in Cancer

The expression of PAR1 is shown to be significantly elevated in various cancers such as breast, lung, pancreatic, colon, gastric, prostate, liver, renal cancer, etc. [88]. On numerous occasions, the coagulation protease-driven activation of PAR1 is shown to promote tumor progression in various ways. The coagulation proteases that are known for promoting tumor progression via PAR1 are aPC, thrombin, and FXa. aPC triggers human breast cancer migration via the EPCR-dependent activation of PAR1 [76]. In gastric cancer, thrombin-induced PAR1 activation triggers tumor growth and invasion via the activation of the NF-ĸB and EGFR pathways [10]. In pancreatic ductal adenocarcinoma (PDAC), thrombin-stimulated PAR1 activation is shown to stimulate the ERK-dependent induction of macrophage chemokine MCP1 and matrix metalloproteinase MMP-9, resulting in PDAC growth and metastasis [77]. In nasopharyngeal carcinoma, the thrombin-induced activation of PAR1 leads to the induction of MMP-2 and MMP-9, triggering tumor metastasis by facilitating extracellular matrix (ECM) degradation and the disruption of the basement membrane [78]. In another study, thrombin-triggered PAR1 activation is observed to stimulate EGFR-dependent p21-activated kinase (Pak1) activity, which promotes the invasiveness of inflammatory breast cancer [79]. Thrombin/PAR1 signaling is also shown to influence tumor angiogenesis in an indirect mechanism. The increased expression of thrombin is observed in the tumor microenvironment (TME) [89] and PAR1 is ubiquitously expressed in various cells in the TME, such as endothelial cells, platelets, macrophages, fibroblasts, etc. [90]. Thrombin-induced PAR1 activation in fibroblasts and endothelial cells augments the expression of pro-angiogenic factors, vascular endothelial growth factor (VEGF), and its receptor, VEGFR2, which could trigger tumor angiogenesis [80]. Thrombin also enhances the endothelial expression of PAF, IL-6, and IL-8 [91], which could promote endothelial proliferation and angiogenesis. Like thrombin, FXa also promotes melanoma growth, which is believed to be dependent on PAR1, as PAR1 agonists show similar effects to that of FXa [81], and FXa is well-known for the activation of PAR1 [34,40]. The perturbation of FXa-driven PAR1 signaling in mice by the administration of FXa-specific inhibitors, rivaroxaban or edoxaban, is also shown to suppress the proliferation of colorectal tumors while accelerating tumor apoptosis [82].

### 3.2. Coagulation Protease-Driven PAR2 Signaling in Cancer

Similar to PAR1, PAR2 is also widely expressed in various cancers [92]. Coagulation protease FVIIa is often shown to promote cancer progression in various ways upon binding to its principal receptor, TF, and the subsequent activation of PAR2. For example, in colon cancer, TF/FVIIa/PAR2 signaling stimulates the proliferation and migration of cancer cells via the PKCα- and ERK-dependent transcriptional activation of c-Jun/AP-1 [83]. In breast cancer also, the TF/FVIIa-mediated activation of PAR2 is shown to promote cancer invasiveness via the AKT/GSK3β-driven nuclear accumulation of β-catenin and the induction of EMT [84]. In another study, the same group reported the contribution of the AKT/NF-ĸB signaling axis to the TF/FVIIa/PAR2-dependent metastasis of human breast cancer via the induction of MMP-2 [11]. PAR2-mediated human breast cancer progression also occurs through indirect mechanisms. TF/FVIIa/PAR2 signaling induces the release of nano-sized microvesicles (MVs) from human highly metastatic breast cancer cells [93], which are enriched with microRNA221 (miR-221), and the MV-mediated transfer of miR-221 confers proliferative, metastatic, and anti-apoptotic potential to non-metastatic breast cancer cells via the induction of EMT [85,94]. PAR2 activation via another protease, trypsin, also induces pro-metastatic MV generation from metastatic breast cancer cells, imparting metastatic potential to non-metastatic breast cancer cells [95]. Like FVIIa, FXa is also shown to stimulate colon cancer growth via the activation of PAR2 and the subsequent intracellular activation of ERK, p38, and AKT [86].

Although the role of PAR1 and PAR2 is well-established in cancer, the contribution of other PARs (PAR3 and PAR4) to tumor pathogenesis remains to be completely understood. However, indirect evidence indicates the crucial role of PAR4 in the thrombin-induced proliferation of human colon cancer cells [87].

## 4. Cancer and Immune Evasion Mechanisms

Immune evasion is a mechanism by which normal cells within the body overcome the attack of the body’s own immune system. Cancer cells are antigenic by nature and are, therefore, vulnerable to recognition by the immune system under normal circumstances. However, cancer cells possess some extraordinary features by which they bypass the immune attack of the body, aiding in their survival. Cancer cells evade host immune response by various mechanisms. The present section briefly discusses the known mechanisms by which cancer cells evade immune attack in the body (Figure 3). In the process of killing cancer cells [96,97,98,99,100], neoantigens released during the death of cancer cells are taken up by the dendritic cells (DCs), which present the antigens to the T-cells through major histocompatibility complex (MHC) molecules, MHC class I and II. Effector T-cells recognize these antigens and get activated. These activated T-cells then infiltrate into the tumor and bind to the cancer cells, resulting in the killing of cancer cells via granule/exocytosis mechanism or apoptotic ligand/receptor interaction. However, defects in the above-mentioned mechanisms often result in bypassing the host’s immune attack against the cancer, leading to the successful development of tumors. Cancer cells evoke several mechanisms to avoid host immune attack (Table 3), as discussed below:

### 4.1. Down-Regulating Immunogenicity of Tumor

On numerous occasions, it has been shown that cancer cells are devoid of immunogenic antigens or typically remove the immunogenic antigens to bypass the immunosurveillance of the host [98,101,102].

### 4.2. Interfering Maturation of Dendritic Cells

Cancer cells often down-regulate the maturation of DC via the release of macrophage colony-stimulating factor (MCSF) [103], IL-10 [104], prostaglandin [105], VEGF [106], TGF-β [107], and indoleamine 2,3 dioxygenase (IDO) [108].

### 4.3. Down-Regulating the Activity of T-Cells

Besides antigen recognition, the complete activity of T-cells essentially requires co-stimulatory interactions between T-cells and DCs, such as CD70:CD27, B7.1/B7.2: CD28, OX40L:OX40, GITRL:GITR, and 4-1BBL:4-1BB. Cells in the tumor microenvironment are shown to express reduced levels of co-stimulatory molecules. T-cell activation without co-stimulation leads to the expression of negative modulating factors, rendering T-cells unresponsive [109,110,111,112].

### 4.4. Perturbation of T-Cell Infiltration

Cancer cells inhibit the infiltration of T-cells in several ways. During T-cell activation and in response to IFN-γ, T-cells express surface chemokine receptors such as CXCR3 [122]. The ligands for the receptor, such as CXCL9, -10, and -11, are expressed in cancer cells, and this receptor-ligand interaction facilitates the infiltration of T-cells into the TME. Cancer cells often bypass this mechanism by either reducing the expression of the ligands, decomposing the ligands, or inducing post-translational modifications, limiting the infiltration of T-cells into the TME [113]. Moreover, tumor cells release VEGF, which targets endothelial cells, leading to the reduced expression of adherent factors, ultimately perturbing T-cell adhesion on vascular endothelium, essential for the infiltration of T-cells [114,115]. Furthermore, tumor cells are shown to secrete IL-10 and prostaglandin E2 (PGE2), which induce the expression of death mediator Fas ligand (FasL) in endothelial cells, thereby leading to CD8^+^ T-cell apoptosis [116]. In addition, cancer-associated fibroblasts (CAFs) are believed to produce ECM components, such as collagen, which are deposited in the tumor substrate, further inhibiting the migration of T-cells towards the cancer cells [117,118].

### 4.5. Inhibition of Immune Recognition

Cancer cells often modulate certain self-molecules that are essential for immune recognition, thereby evading immune response. For example, the expression of surface antigens, MHC-I, proteasome components, β2-microglobulin, TAP1/2, etc., are shown to be down-regulated in cancer cells via the modulation of gene expression, contributing to immune evasion [119].

### 4.6. Up-Regulating the Function of Immunosuppressive Cells

In TME, macrophages are differentiated into M2-phenotypes, which suppress CD8^+^ T-cell response via the release of IL-10 [120]. Myeloid-derived suppressor cells (MDSCs) in the TME are shown to up-regulate the function of regulatory T-cells (T_reg_ cells) while down-regulating cytotoxic T-cell response. MDSC-released TGF-β suppresses cytotoxic T-cell activity via down-regulating the expression of perforin and granzyme [120]. On the other hand, the population of MDSC-induced T_reg_ cells in the TME is shown to be significantly up-regulated, which inhibits the CD8^+^ T-cell response, leading to tumor progression [120]. Cancer cell-expressed IDO induces kynurenine, which suppresses the function of cytotoxic T-cells while inducing T_reg_ cells and MDSCs [121].

## 5. Coagulation Protease-Driven Cancer Immune Evasion

In the previous section, we demonstrated how coagulation proteases influence the progression of cancer in various ways. However, the role of coagulation proteases in cancer immune evasion contributing to disease progression is only just emerging. A few studies indicate that clotting protease thrombin, FVIIa, and FXa contribute to cancer immune evasion via unique mechanisms. The present section briefly illustrates how different clotting proteases influence the progression of tumors by evading a host’s immune responses.

### 5.1. The Role of Thrombin in Cancer Immune Evasion

TGF-β1 plays a crucial role in the immunosuppressive responses of cancer. It suppresses the cancer immune response by not only inhibiting the clonal expansion of cytotoxic T-cells but also reducing their cytotoxicity [123]. TGF-β1 is found to be associated with glycoprotein A repetitions predominant (GARP) on platelets in a latent form (LTGF-β1). The level of thrombin is shown to be significantly higher in patients suffering from cancer-associated thrombosis. Thrombin is shown to cleave GARP on platelets, leading to the release of active TGF-β1, which could promote immunosuppressive functions [124]. Thrombin-induced PAR1 signaling is also observed to suppress the anti-tumor immunity of pancreatic ductal adenocarcinoma, leading to tumor growth and progression [125]. The thrombin-mediated activation of PAR1 in pancreatic ductal adenocarcinoma also triggers the evasion of cytotoxic T-cells via the induction of immunosuppressive genes *Csf2* and *Ptgs2* [126]. The thrombin-induced release of IL-6 from different cells in the TME [127] down-regulates the differentiation, maturation, and antigen-presenting abilities of DC through STAT3 signaling, thereby evading tumor immune response [128]. Thrombin also triggers the release of TNF-α from monocytes, adipose cells, vascular smooth muscle cells, and monocyte-derived macrophages [127], which facilitates tumor immune evasion in multiple ways: it (1) promotes the accumulation and activity of the negative regulatory cells of the tumor immune response, such as T_reg_ cells [129], MDSCs [130], and regulatory B-cells (B_reg_ cells) [131], (2) interferes with the tumor infiltration of cytotoxic T-cells [132,133], and (3) induces the activation-triggered death of cytotoxic T-cells [134]. Different cells in the TME also release MCP-1 in response to thrombin [127], which plays immunosuppressive roles via inducing macrophage polarization into M2 phenotypes, leading to tumor immune evasion [135]. Figure 4 illustrates how thrombin triggers the immune evasion of cancer in different mechanisms.

### 5.2. FVIIa in Cancer Immune Evasion

Although the role of FVIIa is well established in cancer growth and metastasis, its contribution to cancer immune evasion remains understudied. In a recent study, Paul et al., for the first time, showed that the TF/FVIIa-dependent activation of PAR2 triggers the immune evasion of breast cancer via the induction of programmed death-ligand 1 (PD-L1) expression and its stability [136]. PD-L1 is shown to be expressed in various cancer cells, and its receptor, PD-1, is found in the tumor-infiltrating lymphocytes. The interaction of PD-L1/PD-1 not only triggers the down-regulation of lymphocyte proliferation but also induces lymphocyte apoptosis, leading to tumor immune evasion [137,138]. In their study, Paul et al. demonstrated that the TF/FVIIa-mediated activation of PAR2 promotes the expression of PD-L1 in human triple-negative breast cancer (TNBC) cells, leading to a down-regulation of CD8^+^ T-cell activity [136]. The treatment of cells with PAR2 activation peptide shows a similar induction of PD-L1 expression to that of FVIIa and PAR2 knock-down, which significantly down-regulates FVIIa-driven PD-L1 expression [136]. Mechanistically, the authors indicate that TF/FVIIa/PAR2 signaling triggers the inactivation of LATS1, thereby resulting in the loss of YAP/TAZ phosphorylation, leading to their subsequent localization into the nucleus [136]. The Knock-down of PAR2 or blocking cell surface TF dramatically reduced FVIIa-mediated LATS1 inactivation and the nuclear translocation of YAP/TAZ [136]. The perturbation of this signaling pathway manifests a lower expression of PD-L1 by FVIIa, thereby increasing CD8^+^ T-cell activity [136]. In addition, the authors also demonstrate that besides PD-L1 expression, TF/FVIIa-driven PAR2 activation promotes PD-L1 stability via glycosylation through N-glycosyltransferases, STT3A, and STT3B [136]. In earlier studies, the same group reported that TF/FVIIa/PAR2 signaling promotes the AKT/GSK3β-driven nuclear accumulation of β-catenin, thereby triggering TNBC metastasis [84]. In the present study, Paul et al. delineate that the TF/FVIIa/PAR2-mediated nuclear translocation of β-catenin further promotes the expression of STT3A and STT3B, contributing to PD-L1 stability and the associated TNBC immune evasion [136]. The knock-down of STT3A or- B is shown to reduce PD-L1 stability, thereby increasing the activity of CD8^+^ T-cells [136]. In vivo, mice bearing TF knock-out tumors showed reduced tumor growth and enhanced CD8^+^ T-cell population in tumors, which also promotes the efficacy of anti-PD-1 therapy [136]. Figure 5 highlights the mechanism of the FVIIa-mediated immune evasion of human TNBC.

### 5.3. The Role of FXa in Cancer Immune Evasion

Only a few studies identify FXa as an important contributor to tumor immune evasion. In a study by Graf et al., FXa-driven PAR2 signaling is shown to promote tumor immune evasion [139]. In this study, the authors indicated that FXa, originating from myeloid cells, induces the immune evasion of tumors via signaling through PAR2, and FXa-specific inhibitor, rivaroxaban, shows a synergistic effect with anti-PD-L1 therapy in improving anti-tumor immunity [139]. Consistent with these findings, Haist et al. also showed the improved therapeutic efficacy of immune checkpoint inhibitors while administered with FXa inhibitors in patients suffering from metastatic malignant melanoma [140]. Additionally, FXa/PAR2 signaling in tumor-associated macrophages is discussed in the secretion of immunosuppressive mediators, and FXa inhibition is shown to increase the population of cytotoxic T-cells with tumor-killing ability while reducing the Infiltration by immunosuppressive cells, MDSCs, and T_reg_ in the TME [141]. Figure 6 illustrates how FXa promotes immune evasion via the activation of PAR2.

## 6. Biomarkers for Thrombosis Associated with Immune Checkpoint Inhibitors

A biomarker is defined as a material for which the presence in an organism indicates abnormalities, such as infection or disease [142]. The dynamics of C-reactive proteins (CRPs) are primarily used as a biomarker to determine the treatment responses of immune checkpoint inhibitors associated with venous thromboembolism [143] (VTE) [144]. The treatment of patients with immune checkpoint inhibitors results in an immediate burst of CRP levels within a month, which drops beyond the baseline within 3 months [145,146]. The levels of MDSCs, granulocyte-macrophage colony-stimulating factor (GM-CSF), soluble vascular cell adhesion molecule 1 (sVCAM-1), IL-8, and IL-1 receptor against are shown to be significantly elevated in cancer patients suffering from VTE post immune checkpoint inhibitor challenge [147]. This serves as a predictive biomarker for cancer-associated thrombosis after treatment with immune checkpoint inhibitors. A high level of troponin T (TnT; ≥14 ng/L) in cancer patients’ blood following immune checkpoint inhibitors treatment marks the risk of arterial thrombosis-associated cardiovascular anomalies such as stroke, ischemic attack, pulmonary embolism, heart failure, cardiovascular death, etc. [148]. In another retrospective study, troponin I (TnI) was shown to significantly increase (≥50 ng/L) in the plasma of metastatic cancer patients following treatment with the immune checkpoint inhibitor pembrolizumab, which is an indicator of major adverse cardiac events such as VTE, heart failure, acute coronary syndrome, myocarditis, etc. [149]. Therefore, cardiac troponin level is often considered as a predictive biomarker for cardiac anomalies in cancer after treatment with immune checkpoint inhibitors. Table 4 briefly summarizes the predictive biomarkers in cancer patients associated with thrombosis following immune checkpoint inhibitors treatment.

## 7. Immune Regulators in Clinical Trials: The Translational Significance in Cancer-Associated Thrombosis

More often, the results observed in preclinical studies may not be reflected in humans, and in several instances, a drug showing promising results in preclinical studies proves to be unsafe in humans or becomes ineffective. Therefore, before releasing a drug into the market, clinical trials on human subjects become indispensable. Clinical trials basically comprise four phases: (1) phase I usually involves a small group of patients (15–50 individuals) to determine the dose, toxicity, and side effects of the drug [150]; (2) phase II involves a fairly large number of patients (100–300 individuals), determining the efficacy, optimum dose, and safety measures [151,152]; (3) phase III involves a huge number of patients (thousands of individuals), further confirming the efficacy and safety profiles [153,154], based on which the drug is approved and released into the market; (4) phase IV involves post-marketing analysis [155]. Although several immune checkpoint inhibition-based therapies exist in clinical trials, the US clinical trial database (https://clinicaltrials.gov/; accessed on 8 February 2024) currently lists only two clinical trial studies that involve cancer-associated thrombosis and the immune response. The clinical trial ‘Exploring Cancer-Associated Thromboembolism Prognosis Biomarkers and Polymorphisms (CAT_PB)’ (NCT number: NCT06065592) aims to assess the biomarkers and related polymorphisms in cancer-associated thrombosis and their interaction with immune systems. It also analyzes the effectiveness of combination therapy using targeted inhibitors, such as Palbociclib, along with anticoagulants, such as rivaroxaban. The second clinical trial, ‘Neoadjuvant Pembrolizumab and Axitinib in Renal Cell Carcinoma with Inferior Vena Cava Tumor Thrombus (NEOPAX)’ (NCT Number: NCT05969496) primarily focused on evaluating the combined effect of Pembrolizumab and Axitinib in altering inferior vena cava tumor thrombus burden, thereby decreasing surgical complications, leading to improved survival in patients. The details of the clinical trials (both recruiting and non-recruiting with interventional and observational) involving cancer-associated thrombosis and immune response are shown in Table 5.

## 8. Conclusions and Future Directions

Coagulation and cancer are intrinsically related; hypercoagulation-associated thrombotic complications are frequently observed in numerous cancers. The coagulation protease-driven activation of PAR signaling plays a major role in cancer progression via the induction of cancer growth, proliferation, metastasis, and angiogenesis while down-regulating apoptosis. Emerging evidence indicates that cancer cells evoke unique mechanisms by which they evade host immune attack, thereby contributing to tumor pathogenesis. Although the role of coagulation protease-mediated PAR signaling is well documented in tumor progression through various mechanisms, their contribution to conferring cancer immune evasion is only just emerging. The present review highlights the recent findings to understand how coagulation proteases, such as thrombin, FVIIa, and FXa, promote the immune evasion of various tumors. Thrombin produces immunosuppressive functions by interfering with the infiltration and activity of cytotoxic T-cells and the maturation and differentiation of DCs, inducing the accumulation and activity of MDSC as well as regulatory T- or B-cells. FVIIa, on the other hand, confers immune evasion by down-regulating the activity and survival of cytotoxic T-cells. The immunosuppressive mechanisms of FXa include the down-regulation of cytotoxic T-cells and the up-regulation of MDSCs and T_reg_ cells. The conventional immunotherapeutic mechanisms that have shown promising results in clinical trials against various solid tumors are the use of several immune checkpoint inhibitors that target CTLA-4, PD-L1, and/or PD-1. However, it has been shown that cancer cells often surpass these therapeutic mechanisms by various means, contributing to decreased survival in patients. Therefore, the application of combination therapy involving immune checkpoint inhibitors along with other important targets would serve therapeutic outcomes better. As clotting protease-driven PAR signaling has been recently discovered to evade hosts’ immunosurveillance mechanisms quite efficiently, potential inhibitors of coagulation proteases or PAR, along with immune checkpoint inhibitors in combination, would result in boosting a host’s immune response to fight against cancer. This combination therapy may open a new therapeutic window on the treatment of cancer besides conventional therapeutic means.

## Figures and Tables

**Figure 1 cancers-16-01568-f001:**
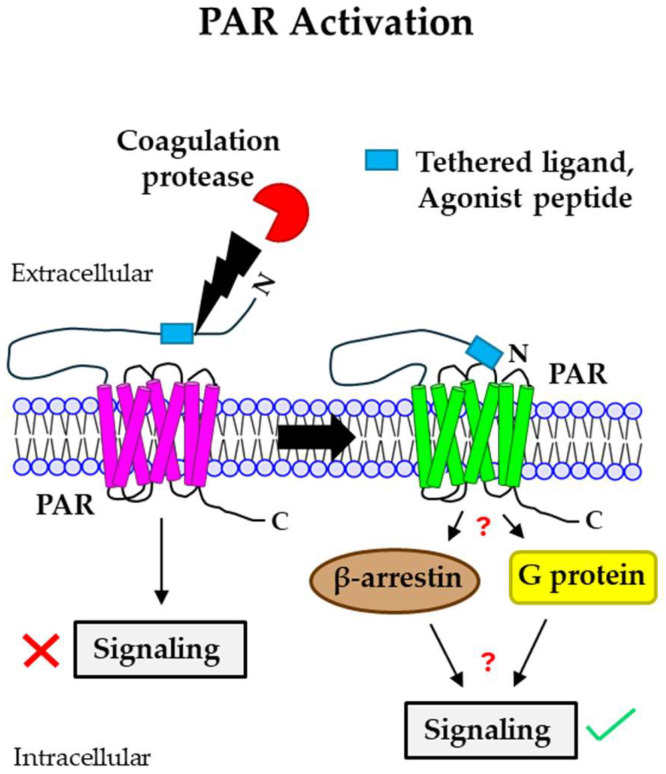
**Mechanism of PAR activation by coagulation proteases.** Coagulation proteases cleave PARs at the N-terminal extracellular domain, leading to the generation of a new N-terminus end. This newly formed N-termini acts as a tethered ligand that binds to the ECL2 region of the receptor itself, resulting in the activation of the receptor. Agonist peptides often activate PARs by directly binding to the receptor and do not require PAR cleavage. PAR activation mediated by different coagulation proteases triggers either intracellular G protein- or β-arrestin-induced signaling, leading to different cellular responses. PAR: protease-activated receptor; ECL2: extracellular loop 2.

**Figure 2 cancers-16-01568-f002:**
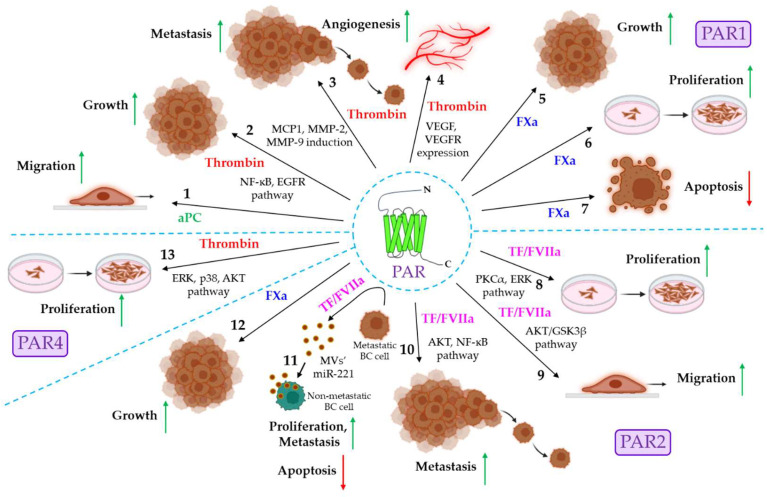
**The role of coagulation protease-driven PAR signaling in cancer.** PAR1, PAR2, and PAR4 are shown to be associated with cancer progression. PAR1 signaling in cancer. **1.** aPC promotes cancer cell migration via PAR1 activation. Thrombin triggers cancer progression in multiple ways. **2.** Thrombin stimulates the PAR1-dependent activation of the C and EGFR pathways, resulting in tumor growth. **3.** Thrombin/PAR1 signaling also induces MCP1, MMP-2, and MMP-9 expression, triggering tumor metastasis. **4.** Thrombin/PAR1 signaling also promotes tumor angiogenesis via the induction of VEGF and VEGFR. **5–7.** The FXa-mediated activation of PAR1 is shown to promote tumor growth and proliferation while down-regulating apoptosis. The role of PAR2 in cancer. **8.** TF/FVIIa/PAR2 signaling promotes cancer proliferation via the activation of PKCα and the ERK pathway. **9.** The TF/FVIIa-dependent activation of the AKT/GSK3β pathway induces the migration of cancer cells. **10.** TF/FVIIa/PAR2 signaling also induces tumor metastasis via the AKT/NF-ĸB pathway. **11.** The TF/FVIIa-mediated activation of PAR2 promotes the release of miR-221-laden MVs from metastatic breast cancer (BC) cells, which deliver miR-221 to non-metastatic BC cells, thereby inducing proliferation, metastasis, and anti-apoptosis to MV-fused recipient cells. **12.** FXa also stimulates PAR2 to promote tumor growth. PAR4 signaling in cancer. 13. Thrombin stimulates PAR4 to induce cancer cell proliferation via the activation of the ERK, p38, and AKT signaling pathways. The green upward arrows indicate up-regulation; the red downward arrows indicate down-regulation. PAR: protease-activated receptor: aPC: activated protein C: NF-ĸB: nuclear factor kappa-light-chain-enhancer of activated B cells: EGFR: extracellular growth factor receptor, MCP1: monocyte chemoattractant protein-1; MMP: matrix metalloproteinase; VEGF: vascular endothelial growth factor; VEGFR: VEGF receptor; FXa: activated factor X; TF: tissue factor; FVIIa: activated factor VII; PKCα: protein kinase Cα; ERK: extracellular signal-regulated kinase; GSK3β: glycogen synthase kinase 3β; miR: microRNA.

**Figure 3 cancers-16-01568-f003:**
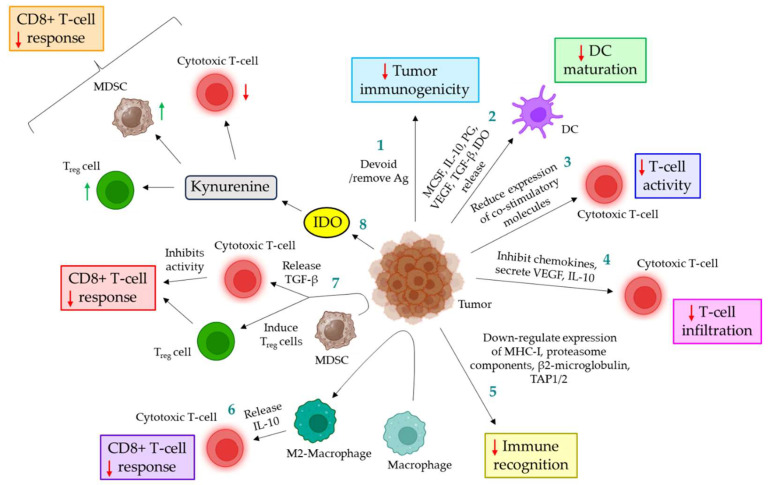
**Different types of cancer immune evasion mechanisms. 1.** Cancer cells are either devoid of antigens or remove antigens, thereby down-regulating tumor immunogenicity. **2.** Tumor cells inhibit DC maturation via the release of MCSF, IL-10, prostaglandin, VEGF, TGF-β, and IDO. **3. **In the TME, cells exhibit lower expression of co-stimulatory molecules, leading to decreased T-cell activity. **4. **Cancer cells inhibit chemokines and secrete VEGF and IL-10, which together down-regulate the infiltration of T-cells. **5. **Tumor cells exhibit a reduced expression of MHC-I, proteasome components, β2-microglobulin, and TAP1/2, thereby avoiding immune recognition. **6. **In the TME, macrophages are converted to M2-macrophages, which promote the release of IL-10, triggering the down-regulation of the CD8+ T-cell response. **7. **In the TME, MDSCs release TGF-β, which down-regulates CD8+ T-cell activity. MDSC also induces T_reg_ cells. Together, these perturb the CD8+ T-cell response. **8. **Tumor cells release IDO, which is converted to kynurenine; this further reduces the activity of cytotoxic T-cells, induces MDSC and T_reg_ cells, and, ultimately, leads to the down-regulation of the CD8+ T-cell response. The Red down arrows indicate down-regulation; the green up arrows indicate up-regulation. MCSF: macrophage colony-stimulating factor; IL: interleukin; VEGF: vascular endothelial growth factor; TGF-β: transforming growth factor β; IDO: indoleamine 2:3 dioxygenase; DC: dendritic cell; VEGF: vascular endothelial growth factor; IL: interleukin; MHC-I: major histocompatibility complex I; TAP: transporter associated with antigen processing; MDSC: myeloid-derived suppressor cells; T_reg_ cells: regulatory T-cells.

**Figure 4 cancers-16-01568-f004:**
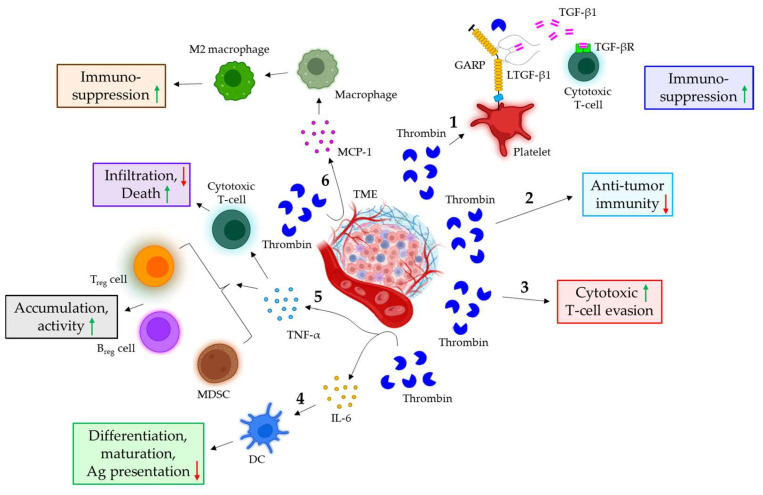
**The role of thrombin in cancer immune evasion. 1.** In the TME, thrombin triggers the cleavage of GARP on platelets, leading to the release of TGF-β1 from the GARP-LTGF-β1 complex. The released TGF-β1 binds to the receptor on cytotoxic T-cells, thereby conferring immunosuppressive functions. **2.** Thrombin also suppresses anti-tumor immunity. **3.** Furthermore, thrombin is shown to promote the evasion of cytotoxic T-cells. **4.** Thrombin stimulates the release of IL-6 from various cells in the TME, which down-regulates the differentiation, maturation, and antigen-presenting ability of DC. **5.** In the TME, thrombin is also shown to induce the release of TNF-α, which not only increases the accumulation and activity of T_reg_ cells, B_reg_ cells, and MDSCs but also evades the cytotoxic T-cell response via down-regulating infiltration and up-regulating apoptosis. **6.** The thrombin-mediated release of MCP-1 in the TME also triggers macrophage differentiation into M2 phenotypes, leading to immunosuppressive responses. The green upward arrows indicate induction; the red downward arrows indicate inhibition. GARP: glycoprotein A repetitions predominant; LTGF-β1: latent TGF-β1; IL: interleukin; DC: dendritic cell; Ag: antigen; TNF-α: tumor necrosis factor α; MDSC: myeloid-derived suppressor cells; T_reg_ cells: regulatory T-cells; B_reg_ cells: regulatory B-cells; MCP-1: monocyte chemoattractant protein 1.

**Figure 5 cancers-16-01568-f005:**
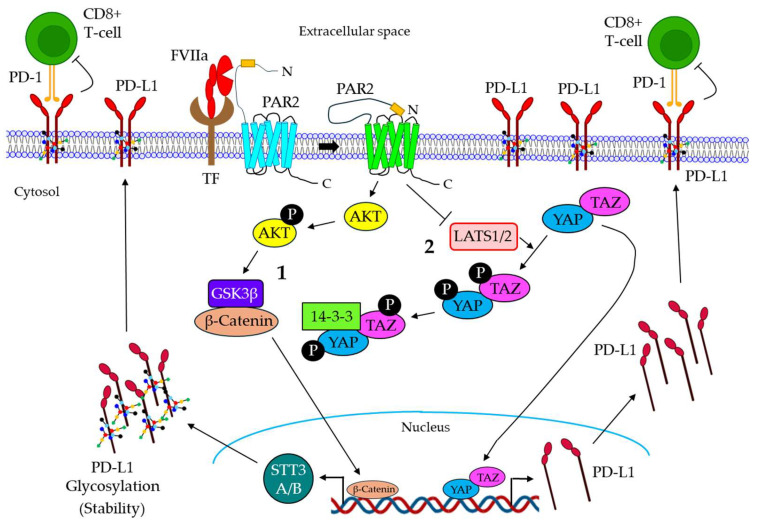
**Mechanism of FVIIa-mediated cancer immune evasion. 1.** The TF/FVIIa-mediated activation of PAR2 promotes AKT phosphorylation, which triggers the GSK3β-dependent accumulation of β-catenin into the nucleus, resulting in the induction of STT3A/B expression in TNBC cells. The induced expression of STT3A/B promotes PD-L1 glycosylation, leading to the stability of PD-L1. **2.** TF/FVIIa/PAR2 signaling also promotes the inactivation of LATS1/2, which is essential for YAP/TAP phosphorylation and their retention in the cytosol. LATS1/2 inactivation results in the nuclear translocation of YAP/TAZ, leading to the induction of PD-L1. PD-L1 is translocated on the surface of the cancer cells, which further binds the PD-1 receptor of CD8+ T-cells, leading to T-cell inhibition. FVIIa: active factor VII; TF: tissue factor; GSK3β: glycogen synthase kinase 3 beta; PD-L1: programmed death-ligand 1; PD-1: programmed cell death protein 1; CD: cluster of differentiation; LATS1/2: large tumor suppressor kinase ½; YAP: yes-associated protein; TAZ: Tafazzin.

**Figure 6 cancers-16-01568-f006:**
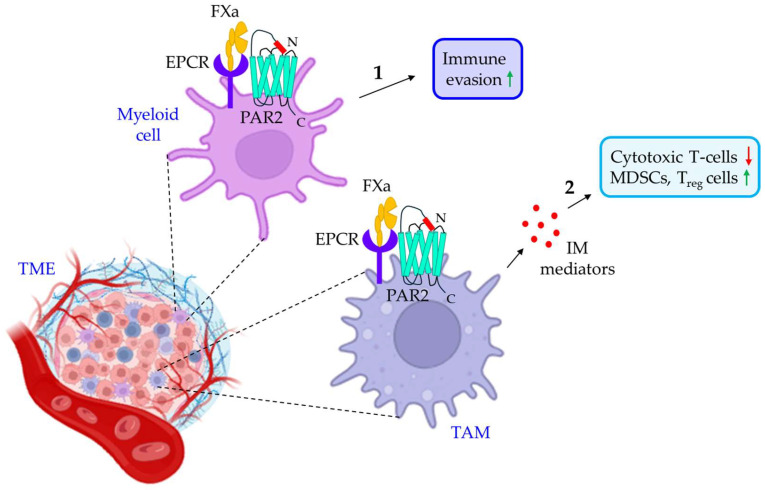
**The role of FXa/PAR2 signaling in cancer immune evasion. 1.** FXa in the TME myeloid cells triggers PAR2 to promote tumor immune evasion. **2.** TAMs in the TME also express PAR2, which is activated by FXa, leading to the release of IM mediators. IM mediators promote tumor immune evasion by either reducing the population of cytotoxic T-cells or up-regulating MDSCs and T_reg_ cells. The green upward arrows indicate up-regulation; the red downwards arrows indicate down-regulation. TME: tumor microenvironment; EPCR: endothelial cell protein C receptor; FXa: activated factor X; PAR2: protease-activated receptor 2; TAM: tumor-associated macrophage; IM: immunomodulatory; MDSCs: myeloid-derived suppressor cells; T_reg_ cells: regulatory T-cells.

**Table 1 cancers-16-01568-t001:** Different PARs, activated by various coagulation proteases and their specific cleavage sites.

PAR Isoform	Cleaving Protease	Cleavage Site	Reference/s
PAR1	Thrombin	R41-S42	[36,37]
	aPC	R46-N47	[38,39]
	FVIIa	R41-S42	[35]
	FXa	R41-S42	[34,40]
	Plasmin	K32-A33, R41-S42, R70-L71, K76-S77, and K82-Q83	[41]
	Kallikrein 14	R46-N47	[5]
PAR2	FVIIa	R36-S37	[42]
	FXa	R36-S37	[42]
	Thrombin	R36-S37	[43]
	Kallikrein-14	R36-S37	[5]
PAR3	Thrombin	K38-T39	[22]
	aPC	R41-G42	[39]
	FXa	R41-G42	[5]
PAR4	Thrombin	R47-G48	[30]
	Kallikrein-14	-	[5]

*Abbreviations:* PAR: protease-activated receptor; aPC: activated protein C; FVIIa: activated factor VII; FXa: activated factor X.

**Table 2 cancers-16-01568-t002:** The role of coagulation protease-mediated PAR signaling in cancer.

Receptor	Cleaving Protease	Cancer Type	Function	Reference
PAR1	aPC	Breast cancer	Promotes breast cancer migration	[76]
	Thrombin	Gastric cancer	Promote tumor growth and invasion via the NF-ĸB and EGFR pathway	[10]
		Pancreatic cancer	Promote tumor growth and metastasis via the induction of MCP-1 and MMP-9	[77]
		Nasopharyngeal cancer	Promotes cancer metastasis via ECM degradation and the disruption of the basement membrane via MMP-2 and -9 induction	[78]
		Inflammatory breast cancer	Induce invasiveness of inflammatory breast cancer	[79]
		Melanoma	Enhances VEGF and VEGFR expression in fibroblasts and endothelial cells in the TME, promoting tumor angiogenesis	[80]
	FXa	Melanoma	Promotes melanoma growth	[81]
		Colorectal cancer	Induces proliferation and prevents apoptosis	[82]
PAR2	FVIIa	Colon cancer	TF/FVIIa/PAR2 signaling promotes colon cancer cell proliferation and migration via the PKCα- and ERK-dependent activation of c-Jun/AP-1	[83]
		Metastatic breast cancer	TF/FVIIa/PAR2 signaling promotes cell migration and invasion via the AKT/GSK3β-driven nuclear translocation of β-catenin and the subsequent induction of EMT	[84]
			TF/FVIIa signaling also promotes metastasis via the AKT/NF-ĸB-mediated induction of MMP-2	[11]
			TF/FVIIa/PAR2 signaling promotes MVs release, which promote EMT to non-metastatic breast cancer cells via the transfer of miR-221, leading to the induction of cell proliferation, metastasis, and anti-apoptosis	[85]
	FXa	Colon cancer	Promotes cancer growth through the intracellular activation of ERK, p38, and AKT	[86]
PAR4	Thrombin	Colon cancer	Induces proliferation	[87]

*Abbreviations:* PAR: protease-activated receptor; aPC: activated protein C; NF-ĸB: nuclear factor kappa-light-chain-enhancer of activated B cells; EGFR: extracellular growth factor receptor; MCP1: monocyte chemoattractant protein-1; MMP: matrix metalloproteinase; ECM: extracellular matrix; VEGF: vascular endothelial growth factor; VEGFR: VEGF receptor; TME: tumor microenvironment; FXa: activated factor X; FVIIa: activated factor VII; TF: tissue factor; PKCα: protein kinase Cα; ERK: extracellular signal-regulated kinase; AP-1: activator protein 1; GSK3β: glycogen synthase kinase 3β; EMT: epithelial to mesenchymal transition; MVs: microvesicles; miR: microRNA.

**Table 3 cancers-16-01568-t003:** Various immune evasion mechanisms exerted by the tumor cells.

Property	Regulation	Mechanism	Reference/s
Tumor immunogenicity	Down	Cancer cells are devoid of immunogenic antigens or remove the antigens	[98,101,102]
DC maturation	Down	Cancer cells inhibit DC maturation via the release of MCSF, IL-10, prostaglandin, VEGF, TGF-β, and IDO	[103,104,105,106,107,108]
T-cell activity	Down	In the TME, cells express reduced levels of co-stimulatory molecules, leading to the expression of negative modulating factors, rendering T-cells unresponsive	[109,110,111,112]
T-cell infiltration	Down	Cancer cells down-regulate the expression of chemokines on their surface or promote them decomposition or induce post-translational modifications, thereby perturbing the binding of chemokines with their receptors on T-cell surface, inhibiting T-cell infiltration in the TME	[113]
	Down	Cancer cell-secreted VEGF targets endothelial cells to inhibit the expression of adhesion molecules, preventing T-cell adhesion on vascular endothelium	[114,115]
	Down	Cancer cell-secreted IL-10 and VEGF triggers the expression of FasL in endothelial cells, leading to apoptosis of CD8^+^ T-cells	[116]
	Down	CAFs are believed to secrete ECM components such as collagen which prevents the migration of T-cells towards the cancer cells	[117,118]
Immune recognition	Down	The expression of surface antigens, MHC-I, proteasome components, β2-microglobulin, TAP1/2 etc. is shown to be down regulated in cancer cells, enabling them to avoid the attack by host’s immune cells	[119]
Function of immunosuppressive cells	Up	In TME, macrophages are differentiated into suppressive cells M2-macrophages which promote IL-10 release, suppressing CD8^+^ T cell response	[120]
	Up	MDSC-released TGF-β down-regulates the expression of perforin and granzyme in cytotoxic T-cells. MDSC-induced T_reg_ cells are highly populated in the TME, inhibiting CD8^+^ T-cell response	[120]
	Up	Cancer cell-expressed IDO induces kynurenine, which suppresses cytotoxic T-cells while inducing T_reg_ cells and MDSCs	[121]

*Abbreviations:* DC: dendritic cell; MCSF: macrophage colony-stimulating factor; IL: interleukin; VEGF: vascular endothelial growth factor; TGF-β: transforming growth factor β; IDO: indoleamine 2:3 dioxygenase; TME: tumor microenvironment; CAF: cancer-associated fibroblast; ECM: extracellular matrix; MHC-I: major histocompatibility complex I; TAP: transporter associated with antigen processing; MDSC: myeloid-derived suppressor cells; T_reg_ cells: regulatory T-cells.

**Table 4 cancers-16-01568-t004:** Biomarkers for thrombosis, associated with immune checkpoint inhibitors.

Biomarker	Cancer Type	Characteristics	Reference/s
CRP	Renal cancer, NSCLC	The levels of CRP rise immediately within 1 month post immune checkpoint inhibitors treatment, which reaches below baseline within three months	[145,146]
MDSCs, GM-CSF, sVCAM-1, IL-8, IL-1 receptor	Lung cancer, melanoma	The levels of MDSCs, GM CSF, sVCAM-1, IL-8, and IL-1 receptor are well elevated in cancer patients with VTE following immune checkpoint inhibitors treatment	[147]
TnT	Squamous cell lung cancer, adenocarcinoma, pleural m esothelioma, neuroendocrine lung cancer	Significant level of TnT in cancer patients’ blood following immune checkpoint inhibitors treatment, which serves as a biomarker for arterial thrombosis-associated cardiovascular diseases	[148]
TnI	NSCLC, renal carcinoma, malignant melanoma, etc.	TnI level in metastatic cancer patients’ blood is elevated following treatment with pembrolizumab, associated with major adverse cardiac events	[149]

*Abbreviations:* CRP: C-reactive proteins; NSCLC: non-small cell lung cancer; MDSC: myeloid-derived suppressor cells; GM-CSF: granulocyte-macrophage colony-stimulating factor; sVCAM-1: soluble vascular cell adhesion molecule 1; IL: interleukin; VTE: venous thromboembolism; Tn: troponin.

**Table 5 cancers-16-01568-t005:** Current clinical trials, according to the US clinical trial database (https://clinicaltrials.gov/; accessed on 8 February 2024), that involve cancer-associated thrombosis and immune response (both recruiting and non-recruiting with interventional and observational).

Trial Name	NCT Number	Characteristics	Cancer Type	Phase	Intervention
Exploring Cancer-Associated Thromboembolism Prognosis Biomarkers and Polymorphisms (CAT_PB)	NCT06065592	Duration: February 2019–December 2024; Population: 500; Age: >18; Sex: M and F	Colorectal cancer	I	Drug: Palbociclib Rivaroxaban Genetic: SNP
Neoadjuvant Pembrolizumab and Axitinib in Renal Cell Carcinoma with Inferior Vena Cava Tumor Thrombus (NEOPAX)	NCT05969496	Duration: December 2023–November 2029; Population: 17; Age: ≥18; Sex: M and F	Renal cancer	II	Drug: Axitinib Pembrolizumab

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
