# Peer review of "Coagulation Protease-Driven Cancer Immune Evasion: Potential Targets for Cancer Immunotherapy"

_cancers, 2024, doi:10.3390/cancers16081568_

Round 1

Reviewer 1 Report

Comments and Suggestions for Authors

Publishing a review article in this field will of great help for those who want to survey current state of cancer-related researches.

Reviewer 2 Report

Comments and Suggestions for Authors

In their review "Coagulation Protease-Driven Cancer Immune Evasion: 2 Potential Targets for Cancer Immunotherapy", Subhojit Paul et al. describe the complex relationships of coagulation activation under the paraneoplastic influence of malignant processes and the possible influencing factors of checkpoint inhibitor therapy.

The comprehensive manuscript is both precise and exhaustively detailed.

The figures are well illustrated, yet very complex in structure, and the legends are well labeled. 

An extensive literature data search has been carried out and the literature sources used have been carefully selected.

The only point of criticism that can be made at this point is that the actual sections on checkpoint inhibitor-addressed therapy are somewhat brief in the overall work.

Overall, the manuscript is a very comprehensive review that I would like to recommend for publication. 

Reviewer 3 Report

Comments and Suggestions for Authors

In this revision the authors carefully revised the relationships of coagulation protease-mediated activation of PAR (protease activated receptors) and its isoforms, concerning their influence on cancer immune evasion. For the work, they listed and reviewed different approaches concerning the complexity of PARs activation farther of its well-known role in tumor progression. Here they explore the contribution of these systems in conferring cancer immune evasion. The authors highlight findings leading to the understanding of how different coagulation proteases operate mechanisms to promote the immune evasion of certain types of cancers. They also discuss the recent immunotherapy approach offering possibility of “application of combination therapy involving immune checkpoint inhibitors along with other important targets would serve better therapeutic outcomes”. Finally, they claim this “combination therapy may open a new therapeutic window in the treatment of cancer besides conventional therapeutic means”.

In my opinion the study-work was well conducted and accordingly described including appropriate illustrations. Sounds conclusions are then made. So, I find sufficient reasons to be in favor this work be accepted for publication.
